# A Qualitative Study on Nudging and Palliative Care: “An Attractive but Misleading Concept”

**DOI:** 10.3390/ijerph18189575

**Published:** 2021-09-11

**Authors:** Ludovica De Panfilis, Carlo Peruselli, Giovanna Artioli, Marta Perin, Eduardo Bruera, Kevin Brazil, Silvia Tanzi

**Affiliations:** 1Bioethics Unit, Azienda USL-IRCCS di Reggio Emilia, 42122 Reggio Emilia, Italy; marta.perin@ausl.re.it; 2Palliative Care Unit, Azienda USL-IRCCS di Reggio Emilia, 42122 Reggio Emilia, Italy; carlo.peruselli@gmail.com (C.P.); giovanna.artioli@ausl.re.it (G.A.); silvia.tanzi@ausl.re.it (S.T.); 3PhD Program in Clinical and Experimental Medicine, University of Modena and Reggio Emilia, 41121 Modena, Italy; 4Department of Palliative, Rehabilitation, & Integrative Medicine, UT MD Anderson Cancer Center, Houston, TX 77030, USA; ebruera@mdanderson.org; 5School of Nursing and Midwifery, Queen’s University Belfast, Belfast BT9 7BL, UK; k.brazil@qub.ac.uk

**Keywords:** palliative care, oncology, nudging, ethics

## Abstract

The debate about the ethical decision-making process in the healthcare context has been enriched with a theory called “nudging”, which has been defined as the framing of information that can significantly influence behaviour without restricting choice. The literature shows very limited data on the opinion and experience of palliative care healthcare professionals on the use of nudging techniques in their care setting. The aim of this study is to explore the beliefs of experienced palliative care professionals towards nudging. We performed a qualitative study using textual data collected through a focus group. It was audio-recorded, and the transcripts were subjected to a thematic analysis. It was performed within an oncological research hospital with a small and multidisciplinary group of healthcare professionals specialised in PC. Participants reported two overarching positions grounded in two main themes: (1) translating nudging in the PC setting and (2) towards a neutral space. The participants found few justifications for the use of nudging in the PC field, even if it can be very attractive and reassuring. Participants also expressed concerns about the excessive risk of developing pure paternalism.

## 1. Introduction

The ethical debate on physician–patient relationships revolves around different approaches, such as classical principlism by Beauchamp and Childress [1], relational ethics [2] and virtue ethics [3]. These theories give value to the shared decision-making process [4], even if each method uses a different conceptual framework. Since 1970, these theories have echoed new legislation, educational programmes and research projects with the aim of promoting patient self-determination through funding for the development, testing, and implementation of decision aids [5]. According to these theoretical and ethical frameworks, the emphasis is on individual responsibility and the right to open communication and full disclosure [6].

In recent years, the ethical debate about the decision-making process has been enriched with a novel theory, proposed by Thaler and Sunstein, called “nudging”. Authors define it as “the framing of information that can significantly influence behavior without restricting choice.” [7]. In other words, nudge represents any aspects of choice architecture that can alter people’s behaviours without forbidding any options [8].

Born in the economic and political sphere, the nudging techniques rest on a key concept known as “libertarian paternalism”. It argues that a subject or an institution may, and perhaps has a duty to, in some way support the choices and habits that help people lead good lifestyles [9].

If we translate the concept into the healthcare system, it is possible to argue that libertarian paternalism alludes to a form of “unobtrusive” intervention that would leave the freedom of the final decision to the individual. Namely, libertarian paternalism helps people in making decisions within a predetermined framework that changes the choice architecture for decision-making [9,10] and “nudges” people into making a “right” choice. Much has been written about the ethics of nudging in adult patients, especially with regards to informed consent issues [11,12,13].

Regarding the field of palliative care (PC), the literature is quite controversial [14,15,16,17,18]. Some authors have tried to apply nudging in the PC setting [14]. They argue that using open-ended questions and offering wide choices to PC patients, especially the ones defined as patients with maladaptive coping, can cause unnecessary suffering to them and their family members [15]. Maladaptive coping is typical of a large range of patients: patients with rigid or limited coping skills, patients who belong to cultural groups mistrustful of the medical community based on historical events and patients with a history of substance abuse. It also applies to patients with serious mental illness or a personality disorder [15].

Conversely, the Covid pandemic has highlighted how much communication with patients and families regarding goals of care became extremely important due to the limited number of inpatient and ICU beds available, as well as the need to reduce aggressive escalation in care close to the end of life [16,17].

To avoid strong emotional reactions and an increase in this patient suffering, poor medical care and complicated bereavement for loved ones, some authors propose “palliative paternalism” as a way to communicate, using few open questions and presenting well-documented and real-choice options during care relationships [15]. On the contrary, other authors [18,19,20] state that nudging is incompatible with honest communication as the core of PC [18]. In adopting nudges, the healthcare professional’s (HP’s) aim is to make it more likely that the patient’s decision is the one that the doctor desires [19,20].

The literature shows very limited data on the personal and professional opinion and experience of PC HPs regarding the use of nudging techniques, and to our knowledge, no study has been conducted on this topic among Italian PC settings.

This qualitative study aims to explore the beliefs of experienced Italian PC clinicians towards nudging. It is part of a larger research project called “Teach for ethics in Palliative care” [21] regarding the implementation and evaluation of an educational programme for HPs working with patients with palliative care needs.

## 2. Materials and Methods

### 2.1. Design

To fully understand the opinions and experiences of a group of HPs working in the Palliative Care Unit and Psycho-oncology Service at the Local Health Service “AUSL-IRCCS of Reggio Emilia”, we chose a qualitative research design, namely a content analysis using textual data collected through a focus group [22,23]. FG is a group discussion related to specific issues and with a specific aim focusing on participants’ interaction [24]. 

Study procedures and reporting followed the Consolidated Criteria for Reporting Qualitative Research (CoreQ) guidelines [25].

### 2.2. Participants

Participants included HPs specialised in PC working in an oncological research hospital. This is a 900-bed hospital, accredited as a Clinical Cancer Institutes (OECI), inside the local health service “AUSL-IRCCS of Reggio Emilia”. HPs were recruited from the Palliative Care Unit (PCU) and the Psycho-oncology Unit. The PCU is a specialised hospital-based service with no beds. Its mission is to perform clinical, training and research activities in PC. At the time of the FG, it included two senior physicians and two advanced practice nurses. The Psycho-oncology Unit cooperates with the PCU by holding clinical consultations and taking charge of PCU staff training. One PC senior physician worked at the PCU as a research supervisor.

Recruitment was conducted by a researcher (LDP), PhD, a researcher in medical ethics and the head of the Bioethics Unit. FG participants were informed by LDP about the objectives of the FG and the type of participation required. One week before the FG, the participants were asked to individually read several articles regarding nudging provided by the research group [7,8,10,15,18,26]. At the beginning of the FG, LDP explained the FG aims and assured participants about content confidentiality.

### 2.3. Data Collection

The study employed a qualitative approach using transcribed textual FG data and thematic analysis techniques [23]. The FG was held at a meeting room of the HPs’ workplace and was audio-recorded. The date and time of the meeting were agreed upon with the facilitator and were compatible with the practice obligations of the FG participants. The interaction between participants was simulated using guiding questions (Table 1). The FG was conducted by a facilitator (LDP) and an observer (GA), an expert in qualitative research. The facilitator presented and guided the discussion and interactions among participants concerning their opinions and experiences in relation to the themes. The observer supported the facilitator to guarantee the internal consistency of the FG.

### 2.4. Data Analysis

We followed the thematic analysis by Braun and Clarke [23]. The analyses involved two researchers (GA) and MP, a PhD student in clinical and experimental medicine, who independently analysed the FG transcript by repeatedly reading the text, extrapolating the themes that emerged and grouping and/or dividing the themes into categories of content. Through an iterative process during the analysis, the researchers verified that, from time to time, the main themes and categories of content that comprise them were consistent with the transcribed FG session and identified significant sentences that condensed and represented the meaning of the themes and identified categories. As the analysis proceeded, the researchers were able to combine an inductive approach (in which themes and categories are derived solely from the data, i.e., from the transcripts) with a deductive approach (in which the categorisation process is structured based on the themes and categories of content identified from time to time). The two categorisations were compared, and the identified differences were discussed until an agreement was reached between the researchers, who proceeded to draw up the definitive categorisation, identifying and describing the extrapolated themes and categories that comprised them. The methodological rigour of the analysis was further guaranteed by the supervision of a third researcher external to the study, an expert in qualitative research. Data will be presented reporting participants’ quotations. Every quotation will be identified by a code, representing the participant speaking and the related number of the meaning unit.

## 3. Results

### 3.1. Participants

The FG session lasted approximately 80 min. Seven persons were involved in the FG; no one who was approached refused to participate in the FG. Participant professional characteristics are described in Table 2. 

### 3.2. Findings

The analyses revealed two fundamental themes: 

1. translating nudging in the palliative care setting—something attractive but dangerous

2. towards a neutral space.

#### 3.2.1. Translating Nudging in the Palliative Care Setting: Something Attractive but Dangerous

Participants defined nudging as a “misleading concept” (c.1.2), potentially “very dangerous”, because it “shows itself as something proposed in the best interest of the patient while on the contrary, it can develop in pure paternalism” (c.2.3). One participant considered it even “diabolically beautiful”, because “it hides from you the true part of what it really affirms” (c.4.2).

The participants found the concept of nudging concerning, as it promotes the belief among those who employ this approach that “they have in their own hands the ability to understand what is in the best interest of the patient” (c.5.2) or that they “presume to know what is right (c.6.1)” for another person.

Nudging was also defined as “very attractive” (c.6.2) and “reassuring” (c.7.2): attractive because “it allows you to save time” (c.6.2) and because this kind of a “gentle push” “can be an anchor to maintain a benevolent paternalism (c.4.3)”, which can also be reassuring for healthcare professionals. After all, “acting in the best interest of the patient” (c.7.2) is reassuring for the physician who does not perceive something wrong when using “techniques that wash away, even just a little, their moral consciousness” (c.7.2).

Participants also recognised that “in clinical practice it is hard not to do nudging” (c.3.4) and that healthcare professionals should pay attention to the fact that if “we are pushing, even just a little bit, we are imposing something” (c.5.4).

During the FG discussion, participants noted the importance of having to pay attention and be very mindful of “the risks of nudging they continually run” (c.7.5) in clinical practice. Thinking about their experience, for example, the participants reported that “how you say something has an influence on the response” (c.2.7) that the patients and their families will give you.

During the FG, it was recognised that often patients “do not know how to relate their own values to the choices they have to make do” (c.3.8), but, on the other hand, when patients have strong preferences and desires, “they do not change their opinion even if you push them to make a certain choice that, from your point of view, is in their best interest,” (c 4.6).

According to participants, “the intrinsic meaning of palliative care approach does not justify nudging” (c.6.8). The FG participants noted that “lots of our patients are totally involved in maladaptive coping” (c.3.9) because of the intrinsic patients’ vulnerability (c.5.9) characterising patients with palliative care needs. It was wrong to give their patients the label of “maladaptive coping” to justify the nudging approach; instead, “we have to support the patients and (…) help them to better manage their situation” (c.4.9).

The participants also found that the concept of “not a competent person” is often ambiguous, as regarding “the quality of one’s own life, there are no incompetent people at all” (c.7.7).

As one of the participants remarked, “often, in the process of communication, the person becomes capable, if supported, of expressing about what matters to them to achieve their personal idea of quality of life. This is what I see every day in my clinical practice” (c.5.12).

Participants affirmed that the real meaning of the palliative care approach is to work with patients to “bring out the specific values of that person, taking into consideration the complexity and the risks that the complexity entails” (c.6.11). Using the nudging technique, there is a risk of “leading the persons towards an end of their life that is my end of life” (c.6.12) as the HPs or that is the current opinion of the society. Finally, the FG participants agreed that “we probably risk, as healthcare professionals, being too focused on obtaining a “good death” for our patient, while on the contrary, we should increase our reflection on the “process”, on “what we do” (c.8,10) to guide them towards a good death.

#### 3.2.2. Towards a Neutral Space

In relation to the opinions reported in some of the articles regarding the difficulty of maintaining an attitude of neutrality when proposing different choices to palliative care patients, the participants admitted that a discussion about nudging could help focus on topics that are of “extreme complexity and great interest (c 4.7)”. They considered it an incentive to have further discussions, such as “if we are to be honest with ourselves, are we really neutral with others? (c.7.13)”, or “what if we, as healthcare professionals, have this attitude (nudging) unconsciously or partially consciously?” (c.8.11).

Neutrality was a concept that participants “can work towards” (c.7.15): this leads them to stay in a “neutral space”. To be considered as such, “neutrality” should provide that “healthcare professionals are aware of their own personal and professional connotations” (c.7.11) and, at the same time, they should also assume that “the patients and their family are invited to do the same” (c.7.12). It emerged how the construction of “neutrality” depends on the way in which these two points of view are “never taken for granted” (c.7.13).

Participants considered being aware of their personal values a way to reach neutrality, as well as to support patients and their families to develop an awareness of their own meanings as well. The FG participants agreed that “when two points of view (even ones very different from each other) meet, a third can arise”, and this third point of view can be considered a “space of neutrality” (c.6.14). 

They concluded that maybe at the end, the “main question regarding nudging is not about the possibility to use it or not but about the best way to really understand what the patient’s true needs are, about maintaining a constant effort to pay attention to the other (6.14).

Participants stated that the HPs’ contribution to the discussion on the goals of care at the end of life is a quest to build an “ongoing exercise of self-reflection” (c.5.10) focused both on the risk, intrinsic in clinical practice, of imposing something on their patients, giving them a “gentle push”, and on the necessity of developing an ongoing training to “foster the patients’ capability to fully understand their own values, helping them make a decision on what they really want” (c.5.12).

The FG participants reported that “an interprofessional framework can help us a lot, but it is currently a very rare opportunity” (3.10).

## 4. Discussion

This paper aimed to explore the beliefs of experienced PC HPs towards nudging. The literature shows very limited data on the personal and professional opinions and experiences of HPs regarding the use of nudging techniques, especially in a PC setting.

According to participants, nudging arguments do not help in ensuring an informed and conscious decision or in raising awareness of the patient’s values and personal beliefs.

On the contrary, the FG results highlight a definition of autonomy as “relational”. Dignity, respect, empathy and care are key concepts in the definition of relational autonomy [1,27,28,29,30,31,32,33,34]. Criticising the classical concept of autonomy, i.e., the idea of individuals who make all the choices that concern them rationally, some thinkers assume that “we cannot expect patients to take on so much of the burden of making choices”; they argue that it is necessary to allow for more in the way of directive or confrontational counselling [35,36,37]. For this reason, HPs must be aware of the fine line between relational autonomy and nudging [38].

The decision-making process regarding patients approaching their end of life is a complex topic, as also highlighted by the FG participants [13,39]. Following the study results, it is possible to identify the following four different models to describe relationships of care with patients approaching their end of life. Figure 1 describes the relationship between the models identified (Figure 1: Relationship of care and end-of-life care).

1. Paternalism: The physician consciously orients the patient to give their consent to treatment choices that they (the doctor) consider the right ones for the patient in that situation, regardless of their (the patient’s) values and wishes.

2. Nudging: The physician influences the patient’s choices in a “gentle” manner and through techniques that do not reduce the theoretical possibilities of free choice. While this technique pays more attention to the patient’s wishes than pure paternalism, the physician steers the patient towards the solutions considered more correct, considering that in a situation of severe fragility (as often happens in the conditions of approaching the end of life), the patient’s autonomy is precarious and unreliable. From this point of view, nudging can be considered between a paternalistic framework and a relational autonomy attitude, which can finally be considered as palliative paternalism.

3. Respect for the principle of patient autonomy in a relational context: The care relationship remains focused on respecting the patient’s values and wishes, even if in an inevitably relational context, and puts quality of life at the centre according to the perspectives of the patient with the aim of helping them to make autonomous choices.

4. Absolute respect for the principle of patient autonomy: after having informed the patient about different therapeutic alternatives, the physician lets the patients make the decision independently, running the risk of leaving them alone and faced with often complex and difficult choices.

According to the FG participants, the boundary between the use of nudging techniques and the respect for patient autonomy in a relational context is potentially weak, especially in end-of-life discussion. Then, the PC team has to maintain a moral responsibility towards the care pathway through an ethical approach aiming at helping people answer the question: “What is best for me?”. It is crucial to show patients that their free choices can be empowered by increasing awareness. At the same time, PC HPs should develop an ethical sensitivity in their daily practice.

## 5. Conclusions

The principles of nudging techniques—in the form of a palliative paternalism—at the end of life can present several critical issues and some substantial risks. Some of the theoretical reasons proposed in support of “palliative paternalism” such as the concept of maladaptive coping or the difficulty for some persons to give a motivated judgement have to be critically considered.

An ethics based on patient care needs to be grounded in the patient–physician relationship, making it necessary to rely on the physician’s moral sensitivity [40]. HPs can recognise patients’ wishes and preferences, but their capacity for compassion, honesty, integrity and sense of humility is equally important [40]. 

These qualitative findings can represent the basis to incorporate the ethical skills in communication training programmes at an international level and to foster the implementation of applied ethics courses in Italian PC educational programmes. However, more research is needed to better characterise the role of this challenging intervention for communication and decision-making in PC, explore these qualitative findings with other health professions and understand the role of nudging in PC clinical practice.

## 6. Study Limitations

The most important limitation of this study is the small size of the participants from one single centre. There was also a bias related to the participant’s sex due to the composition of the services involved (when the FG was held, just one man was working at the PCU). This decision was taken, as we involved the participants of an ethics training programme to collect data from experienced participants. Another limitation is that the focus group was not video-recorded due to the research project characteristics of which the focus group was part.

## Figures and Tables

**Figure 1 ijerph-18-09575-f001:**
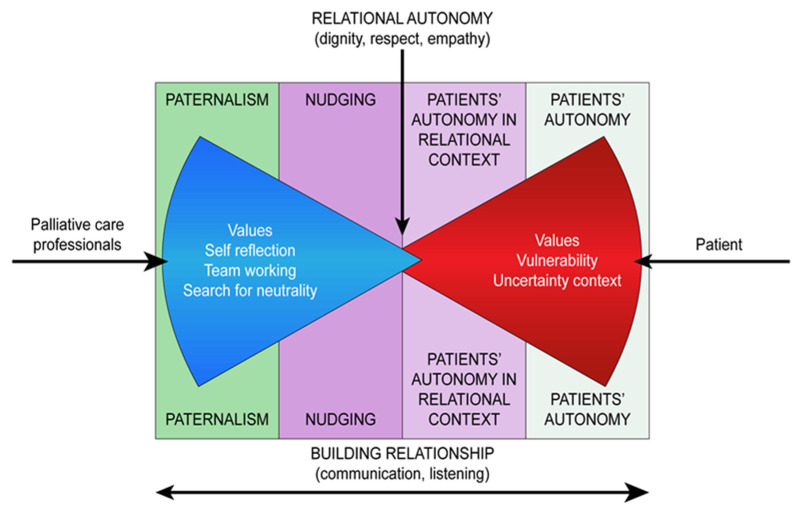
Relationship of care and end-of-life care.

**Table 1 ijerph-18-09575-t001:** The FG guiding questions.

**1**	Do you think that clinical nudging occurs in your daily care relationship?
**2**	Does palliative care approach differ from clinical nudging? If yes, how?
**3**	Are there ways to oppose nudging?

**Table 2 ijerph-18-09575-t002:** Characteristics of the FG participants.

Code	Professional Features	Years of Experience in PC	Sex
1	Specialist in Palliative Medicine	8	Female
2	Specialist in Palliative Medicine	4	Female
3	Nurse in Palliative Medicine	1	Female
4	Nurse in Palliative Medicine	5	Female
5	Psychologist	15	Female
6	Psychologist	10	Female
7	Specialist in Palliative Medicine	30	Male

## Data Availability

All data generated or analysed during this study are included in this published article.

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
