# Peer review of "A Qualitative Study on Nudging and Palliative Care: “An Attractive but Misleading Concept”"

_ijerph, 2021, doi:10.3390/ijerph18189575_

Round 1

Reviewer 1 Report

This is overall a very well written article adressing the important concept of nudging in the context of a patient/health care provider relationship in the palliative care setting.

My main comments are with regards to the methods - I would appreciate the authors expanding their section on study limitations. FG needs to be explicited as "Focus Group" Their focus group is small (7 persons), monocentric; it is unclear whether the group met once or multiple times. The sex ratio of the group is also very biaised; we don't know the size of the local health service in which the providers are working (ie is 7 persons a large enough "sample"?).

One researcher conducted recruitment and facilitated the meeting. The meeting was audio recorded but not video recorded. We nevertheless note that an observer was present to collect non-verbal communication elements.

I would also appreciate more elaboration on perspectives for this work - at the national level in Italy for example. What is needed to confirm the initial qualitative findings from the present study?

Reviewer 2 Report

The manuscript presents the results of a study explored the beliefs of experienced palliative care professionals toward nudging 

 I found the article interesting but I want to highlight some improvements.

If figure 1 is self-made, I think it should be specified whether it is the result of the current study or based on other studies.

I understand that the numbering that appears on lines 247,251,251,254 and 258 is representative of the concepts that appear in the figure. I think it would be clearer to the reader if they appeared in order, i.e., for example, from left to right. In this way, 1: Paternalism, 2: Nudging, 3. Patients´autonomy in relational context, 4. Patients´autonomy.

Thank you very much
